# Investigation of the feasibility and acceptability of a school-based intervention for children with traits of ADHD: protocol for an iterative case-series study

Abigail Emma Russell ,[1] Barney Dunn,[2] Rachel Hayes,[1] Darren Moore,[3] Judi Kidger ,[4] Edmund Sonuga-Barke,[5] Linda Pfiffner,[6] Tamsin Ford [7]

For numbered affiliations see end of article.

**Correspondence to**
Dr Abigail Emma Russell;
A.E.Russell@exeter.ac.uk

## ABSTRACT

**Introduction** Attention deficit/hyperactivity disorder (ADHD) is a prevalent and impairing cluster of traits affecting 2%–5% of children. These children are at risk of negative health, social and educational outcomes and often experience severe difficulties at school, so effective psychosocial interventions are needed. There is mixed evidence for existing school-based interventions for ADHD, which are complex and resource-intensive, contradicting teachers' preferences for short, flexible strategies that suit a range of ADHD-related classroom-based problems. They are also poorly evaluated. In this study, a prototype intervention comprising a digital 'toolkit' of behavioural strategies will be tested and refined. We aim to refine the prototype so that its use is feasible and acceptable within school settings, and to establish whether a future definitive, appropriately powered, trial of effectiveness is feasible. This novel iterative study aims to pre-emptively address implementation and evaluation challenges that have hampered previous randomised controlled trials of non-pharmacological interventions.

**Methods and analysis** A randomised iterative mixed-methods case-series design will be used. Schools will be randomised to the time (school term) they implement the toolkit. Eight primary schools and 16–32 children with impairing traits of ADHD will participate, along with school staff and parents. The toolkit will be refined after each term, or more frequently if needed. Small, theory-based and data driven changes hypothesised as relevant across school contexts will be made, as well as reactive changes addressing implementation barriers. Feasibility and acceptability will be assessed through quantitative and qualitative data collection and analyses in relation to study continuation criteria, and ADHD symptoms and classroom functioning will be tracked and visually evaluated to assess whether there are early indications of toolkit utility.

**Ethics and dissemination** Ethical approval has been obtained. Results will be presented in journal articles, conferences and through varied forms of media to reach policymakers, stakeholders and the public.

## STRENGTHS AND LIMITATIONS OF THIS STUDY

⇒ This study uses an iterative staggered design in order to refine an intervention throughout the research process.
⇒ Measures of acceptability and feasibility of the intervention and of the ability to evaluate it in a rigorous trial setting will be reviewed against continuation criteria throughout the study.
⇒ Multiple informants will provide relevant data, including school staff and parents, and data on both positive and adverse outcomes will be captured.
⇒ However, the iterative nature of the study means that different participants will not receive the same intervention, therefore group-level effects cannot be delineated.
⇒ Alternate analyses will be used to ascertain whether there are patterns of change in symptoms and functioning that replicate across individuals instead.

## INTRODUCTION

Children and young people's mental health and neurodevelopment (including attention deficit/hyperactivity disorder (ADHD)) is a government priority. In the 2017 green paper, the pivotal role of schools in supporting good mental health was acknowledged. By the end of 2023, >20% of schools across the UK will have a designated mental health lead with the aim of improving access to evidence-based treatments.[1] Cost-effective, implementable, sustainable evidence-based mental health interventions that can be delivered in schools are therefore urgently needed.

ADHD is a neurodevelopmental disorder affecting 2%–5% of children, characterised by impairing levels of impulsivity, hyperactivity and/or inattention.[2] Multiple negative life course outcomes are associated with childhood ADHD including high rates of co-occurring mental disorders, accidents,

BMJ

poor educational and occupational attainment and antisocial behaviour.[3] Other neuropsychiatric disorders and unintentional injury are strongly associated with ADHD, which are the leading cause of years lost due to disability worldwide in those aged 10–24 years.[4] Costs to the health, education and judicial systems, social services and economic loss are substantial; 24% of the mean annual cost of £5493 is attributable to National Health Service resources.[5] Pharmacological treatment is available and effective for some, especially in acute management of symptoms, but is not appropriate, acceptable or tolerable for all children.[6] Response is often partial,[7 8] with tolerance developing over several years.[9 10] Even with drug treatment, ADHD causes particular problems in school, resulting in the outcomes described above.[11] Although ADHD medications have been shown to be effective and safe for the treatment of ADHD in the shorter term,[10] licensed ADHD medications are controlled drugs and in order to access these families must receive a clinical diagnosis of ADHD, waiting lists for which in the UK are currently backlogged. Medication is not appropriate or preferable for all children with ADHD, and in the UK is not recommended as a first-line treatment for children with traits that are not severe;[12] and non-pharmacological intervention options that can be used as a precursor, alternative or supplement to pharmacological treatments are important.

ADHD causes problems in the classroom for the child, the teacher and for other children.[13] Symptoms of ADHD (which are also traits present to varying degrees in all children) make it challenging for a child to sit still, be attentive for sustained periods of time, listen to and follow instructions, or to resist impulses to shout out. Children with ADHD traits may also wander around the classroom or school building, struggle to complete schoolwork and need reminders to know what they are doing. This combination of characteristics is challenging for the child themselves; as they are not consciously able to control their behaviour without learning self-monitoring and regulatory strategies and are often criticised by adults and peers for their involuntary behaviour. It can also be disruptive to the usual mainstream classroom context, and there is a poor fit between the UK mainstream Primary classroom environment and the behavioural characteristics and needs of children with impairing traits of ADHD. Development of an effective school-based intervention for ADHD that overcomes the limitations of previous interventions would be likely to improve health and social outcomes. The limitations of existing school-based non-pharmacological interventions are that they usually require delivery by trained clinicians, are complex and multicomponent: targeting many potential deficits in every child regardless of their individual strengths and weaknesses.[14–16] ADHD encompasses a wide range of core associated symptoms and functional issues that are seen in the classroom, and existing evidence is unclear about what works for whom, or which aspects of interventions are effective. In combination, this means existing interventions are not feasible or affordable for schools to routinely deliver; they are rarely rolled out in schools and are not followed with high fidelity. There are barriers to school staff identifying evidence-based practices and implementing these with high fidelity as well as adapting to the needs of the individual, and sustaining this over time. Reviews of existing school-based interventions for ADHD have highlighted the potential efficacy of such interventions,[17–20] and so overcoming the limitations of existing interventions is crucial to provide an implementable, effective school-based intervention for ADHD.

We are in the process of developing an intervention from high-quality evidence, based on theory around behaviour change and ADHD and from a programme of development work including a Delphi survey of key stakeholders. We are following the Intervention Mapping approach[21 22] to co-create the intervention with key stakeholders in order to adapt the format of evidence-based strategies, targeting both symptoms and associated functional impairments, to overcome key implementation issues in schools.[23] Intervention Mapping has six steps, and the full process of the toolkit development will be detailed in a subsequent publication, however in brief these are: (1) Creating a logic model of the problem; (2) Defining programme outcomes and objectives and a logic model of change; (3) Programme design; (4) Programme production; (5) Programme implementation plan; and (6) Evaluation plan.[22] The result will be a 'toolkit' of strategies in a modular format, with some core components then a selection of optional modules focusing on a different core outcome, that is designed for Primary school staff to use when working with children with traits of ADHD: the Tools for Schools *FLEX* toolkit. The toolkit modules target outcomes that were shown to be of importance to key stakeholders in a Delphi survey;[24] including conflict with teachers and peers, paying attention, building self-esteem, and improving planning and organisation.

Given the limitations of existing school-based interventions for ADHD, including limited evidence of effectiveness and difficulty evaluating them in methodologically rigorous randomised controlled trials, it is clear that the toolkit will need to be developed in a way that ensures it is acceptable, useful and feasible for school staff. In addition to this, we must address key features that will allow successful trials of the toolkit in future powered trials so that the strength of evidence for its effectiveness is comparable to other treatments such as medication. Intervention development frameworks such as the Medical Research Council complex intervention guidelines highlight the importance of the development and feasibility phases (rather than prematurely evaluating interventions in large randomised trials); carefully optimising protocols, working closely with the target populations, planning and testing implementation, and assessing the acceptability and feasibility of the toolkit are essential to develop an intervention that can later be tested for effectiveness. The development of the toolkit

will therefore take an iterative case-series approach, trialling and modifying successive versions of the prototype in order to meet the study objectives. We have not included a control group or comparison group as the study does not aim to assess effectiveness.

In this protocol we outline plans to conduct the initial evaluation and further development of the Tools for Schools *FLEX* toolkit. As the toolkit is at an early stage of development, we plan to assess the acceptability of the toolkit, and the feasibility of implementing it within the Primary school context in the UK.

## Objectives

The main aims of this feasibility study are to:
1. Refine the prototype *Tools for Schools FLEX* toolkit so that it is feasible and acceptable to implement in the school setting.
2. Establish whether a future definitive trial of effectiveness is feasible.

Secondary aims are to:
1. Identify suitable outcome measures to assess core ADHD symptoms, child and teacher well-being, academic progress, and identify an appropriate primary outcome (from these) for a future definitive trial.
2. Develop and test a framework for costing the toolkit, and for assessing cost-effectiveness in a future definitive trial.
3. Assess whether observational measures of behaviour, classroom functioning and teacher-reported ADHD symptoms indicate improvement following use of the toolkit.

## METHODS AND ANALYSIS
## Design

A randomised iterative mixed-methods case-series design will be used. Schools will be the units of randomisation, and will be randomised to the time (school term) when they will use the toolkit. The toolkit will be refined after each school has used it for one term, or more frequently if major issues are highlighted during a school's use of the toolkit. Small, theory-based and data-driven changes hypothesised to be relevant across the school context will be made, as well as reactive changes addressing barriers to implementation. Randomisation to using the toolkit in different school terms allows for inferences to be made as to whether the intervention is improving outcomes, with consistent improvement following the introduction of the core components enabling researchers to potentially rule out alternative explanations for behaviour change, such as improvement due to other support in place or differences in behaviour across the school year. If baseline symptoms and functioning remain on stable trajectories for children across schools, changing only when the intervention is introduced, this implies it is the intervention rather than an alternate factor stimulating the change. It is a commonly used design for single-subject analyses

**Figure 1** Case series design outline. Key: grey, baseline; green, intervention; blue, follow-up.

where the intervention cannot be withdrawn and re-implemented such as in an ABAB design.[25–27]

Eight schools will participate in total in two recruitment cohorts of four (in order to avoid later participating schools having to sign up to the study years in advance); each cohort will enter the study at a different time. Schools in each cohort will then be randomly assigned their individual baseline term in the study using a computerised random sequence generator (see figure 1). The length of the baseline period will be the same for each school (one school term), with schools within each cohort entering their baseline period in different school terms; the term in which the intervention is delivered is therefore staggered across the study (figure 1). This design will allow for refining and adapting the toolkit and its implementation based on ongoing participant feedback, maximising the chances it has of being acceptable and feasible (and therefore suitable for further evaluation in a pilot and then definitive cluster randomised controlled trial) by the end of the study.

## Patient and public involvement

A planning group of public and patient representatives, including people with ADHD (adults (n=5) and children (n=9)), parents of children with ADHD (n=12), school staff (n=9), and education and health psychologists (n=4) has been established to co-create the intervention. Input from this planning group has already extensively informed the design of the intervention and study, refining research questions, selecting outcome measures and considering participant burden. The group will also shape study dissemination plans. The planning group contains around 25 people, and will be consulted throughout the course of the study to make decisions about necessary modifications to the intervention, delivery or research design of a future evaluation. In relation to the study objectives, co-developing the intervention with key stakeholders in the planning group increases the chances of the toolkit being feasible and acceptable in the Primary school context. The group will actively collaborate to refine the toolkit in response to participant feedback to achieve the first aim. They will be consulted as to whether

the feasibility study results indicate that a future trial is feasible, in relation to understanding the core features of a high-quality trial. The role of the planning group in co-production of the toolkit is outlined in further detail in the online supplemental material.

Regarding the secondary objectives, the planning group will use data collected to agree on the final primary outcome for a future definitive trial, as well as judge whether trialled measures are suitable measures to assess ADHD symptoms, academic attainment, and child and teacher well-being. The planning group will contribute to the development of the costing framework and evaluate findings from study participants. Finally, the planning group will discuss the findings from observational measures in relation to the toolkit use, helping to understand why any emerging patterns in the data are seen.

### Recruitment and participants

Eight Primary schools, associated school staff and up to 32 children with impairing traits of ADHD (minimum 16 children) will participate in the study, and use the toolkit. All schools and all child participants will receive the intervention. Baseline data regarding treatment as usual will be collected in order to assess normal fluctuations in symptoms and response to any school-implemented treatment as usual during the baseline phase. Baseline data will not be used to exclude participants who have unstable profiles, rather to greater understand the nature of symptom and impairment fluctuation over time.

Recruitment will be through opportunistic sampling, using existing contacts from the planning group such as local Educational Psychology teams, and other education networks in the South West. At the time of writing, four schools have consented to participate and, given the demand for support managing children with ADHD in school, we anticipate no problems recruiting a total of eight schools. This sample of eight mainstream primary schools in the South West of England will allow sufficient variety to purposively sample schools to include a minimum of four in areas of high socioeconomic disadvantage, based on the selection of schools with an index of multiple deprivation above the regional mean. The intervention needs to be able to respond to the range of children with traits of ADHD, including those who are often unrecognised or less likely to access other support, such as girls,[28] therefore eligible schools will have at least one female pupil meeting eligibility criteria and we will aim to recruit a diverse sample of children with differing ethnicities. A sample of eight schools also allows for the intervention periods to be staggered across the study and successive iterations of the toolkit. All participating schools will use the toolkit in the study for one school term.

Study participants include the school's senior leadership or head teacher, who will consent to schools' participation; the school special educational needs (and disabilities) coordinator (SENCo), teachers and teaching assistants (TAs) working with eligible children, eligible children with impairing traits of ADHD, and their parents. Consent will be obtained by a member of the research team through meeting with potential participants face to face or online.

### School and school staff consent

Each school's senior leadership team will consent to the school's participation. They will nominate a mental health lead, who is likely to be the SENCo (as children with ADHD and associated difficulties currently fall under their remit in the UK) and will be referred to as such henceforth. Decisions around recruitment and eligibility criteria have been made in close consultation with the study planning group.

As part of the condition of each school's participation, the senior leadership team will be asked to ensure that the SENCo and potentially eligible teachers and TAs (those working with recruited children) are supportive of and potentially willing to take part in the study.

### Identification and recruitment of eligible children and families

Eligible children will either have a clinical diagnosis of ADHD (as reported by the school or parents), or high symptom levels and will be aged 4–10 years (school years Reception-5) at the first baseline assessment to ensure they will be in primary school for the study duration. Children with ADHD may either have not yet received a formal diagnosis, or their family may not wish to pursue a diagnosis, therefore, children with high symptom levels will be eligible. There are no additional exclusion criteria. Teachers and SENCos will identify students who meet criteria for this indicated group using the Strengths and Difficulties Questionnaire Hyperactivity-Inattention Subscale and the impact supplement.[29] Teachers of potentially eligible children will be asked by the SENCo to complete this brief screening measure, and if this indicates that a child may have probable ADHD (teacher-reported symptoms being ≥6 and impact ≥1), their parent will also be asked to complete the screen in order to ensure that impairments are noted across more than one setting.[2 30] Should these combined ratings indicate 'probable' ADHD according to the validated algorithm, the child will be eligible for inclusion in the study. Probable ADHD is assigned if the parent-report hyperactivity/inattention score is ≥7 and an impact score of 2 is given, or if the hyperactivity/inattention score is ≥9 and an impact score of at least 1 is reported, and the teacher criteria above are met (see sdqinfo.org for syntax).

SENCos will make initial contact with parents of children to clarify eligibility, to provide the study information and to establish whether families are interested in participation in principle. Willing families will be asked to consent to having their details passed to the research team, who will obtain written informed consent from parents and assent from children to participate. Children will be informed of the broad purpose of the research through an age-appropriate information sheet and assent form, unless their parents request otherwise. Should an

**Table 1** Sample size minimum and maximum

| Participant type | Minimum number | Maximum number* |
|---|---|---|
| School | 6 | 8 |
| Children | 16 | 32 |
| Parents or carers | 16 | 64 (ie, 2 parents per child) |
| SENCos | 8 | 8 |
| Teachers | 16 | 32 |
| TAs | 0 | 32 |

*Maximum number based on every eligible school recruiting four pupils, each of which have a different class teacher. We anticipate actual numbers of TAs will be lower than the maximum.
SENCo, special educational needs (and disabilities) coordinator; TA, teaching assistant.

individual child not meet eligibility criteria but their parents and teacher feel they would benefit from participation, this will be considered on a case-by-case basis. As the intervention is embedded as part of usual school practice, the planning group has indicated that ongoing assent should then be assumed for children through data collection unless the child indicates distress or reluctance.

### Sample size
The planning group has advised that to minimise burden of data collection on teachers, each class teacher should have a maximum of one child participant however this may be revised during the course of the study; for example, if a school feels strongly that several children in one class may benefit, or if two eligible children in the same class also have TA support and the teacher wishes both to participate. Estimating that each of the eight participating schools will have 2–4 participating children and teachers, there will be a total of 16–32 child participants, 16–64 parents, and 16–64 teachers and TAs along with eight SENCos. The total sample size will depend on the distribution of participating parents, teachers and TAs and the distribution of eligible children across different classes within schools: table 1 indicates the minimum and maximum sample sizes.

Given that the aim of the study is not to assess effectiveness or efficacy, the potential for the intervention to change between participants, and the case-series design, this study will not be powered to conduct a conventional between-individuals analysis. Instead, we will use multiple individual measurement points to analyse within-individual change and assess whether this replicates (at a descriptive level) across individuals. The pragmatic choice of eight schools allows for this purposive sampling, and both the minimum and maximum sample size of children is sufficient to address our objectives. Based on the mean number of children in eligible year groups in Devon schools, and a conservative estimate of prevalence of ADHD in this age group of approximately 3%, it is expected that each school will have four to eight eligible children, of whom we hope to recruit at least 50%. The recruitment target is feasible: school staff have indicated

**Table 2** Study progression criteria

| | Green: acceptable | Amber: discuss, modify | Red: redesign | Assessing |
|---|---|---|---|---|
| Recruitment of schools | 6 or more | 5 | 4 or fewer | Feasibility |
| Recruitment: teachers, children, parents (considered as separate groups) | >65% | 20%–65% | <20% | Feasibility |
| Retention of schools, teachers, children and parents in study | >65% | 40%–65% | <40% | Feasibility |
| Training completed (teachers) | >90% | 70%–90% | <70% | Acceptability and feasibility |
| Introductory video watched (parents) | >50% | 20%–50% | <20% | Acceptability |
| Child strengths activity completed | >50% | 20%–50% | <20% | Acceptability |
| Adherence to digital Daily Report Card | >70% | 50%–70% | <50% | Acceptability |
| Teacher-completed measures | >70% | 50%–70% | <50% | Feasibility |
| Parent-completed measures | >50% | 20%–50% | <20% | Feasibility |
| Child-completed measures | > 70% | 50%–70% | <50% | Feasibility |
| Observational measures | >70% | 50%–70% | <50% | Feasibility |
| Attendance at toolkit-related meetings | >75% | 40%–75% | <40% | Acceptability |
| Percentage of occasions toolkit reportedly used as instructed | >75% | 50%–75% | <50% | Acceptability and feasibility |
| Follow-up qualitative data collection completed (for each type of respondent: school staff, parents, child) | >50% | 20%–50% | N/A | Feasibility |

that a toolkit for ADHD would be perceived as useful and not burdensome as they believe it would be beneficial to their wider classroom management (the planning group highlight the data collection burden as being the reason that teachers should only have one participating student per class).

An established set of study progression criteria are shown in table 2 and will be used to assess success in terms of meeting the study aims. Criteria relate to the recruitment of schools (five or more being acceptable), recruitment of parents and teachers, and completion of study measures as well as some basic metrics of fidelity. The criteria will be reviewed throughout the study, and where individual criteria are Amber or Red, a core academics team meeting and a planning group meeting will be held to discuss, agree and then implement modifications to the study procedure or content.

## Procedure

The study will run from September 2022 until January 2025. Data will primarily be collected at the participating schools, however interviews with parents and school staff may take place over the phone, via internet video call, at participants' homes, or in another location of the participant's convenience. Baseline length will be one school term. During the baseline phase, the teacher will provide repeated measures of the primary quantitative outcomes: participating children's ADHD symptoms and classroom functioning. These will be collected every 2 weeks during the baseline period and in the intervention period. This detailed baseline data will capture normal fluctuations in symptoms across the term and response to any school-implemented 'treatment as usual' during the baseline phase.

Information on healthcare and education resource use will be collected from children, parents and school staff for each child during the baseline period using a structured survey with open questions to capture any resources not mentioned, and child quality of life will be reported by the child and their parent. Any other support or treatment will not be altered during the study. Each school will then implement the intervention for one school term; two modules from the intervention will each be implemented for 4 weeks. There will be a 10-week follow-up period where detailed qualitative data will also be captured, including asking participants about changes in resource and service use relative to the information they provided at baseline.

## Intervention delivery

The intervention is currently in development. Broadly, it will take the form of a digital toolkit of training and resource packages and behavioural strategies for non-specialists to use, organised within 'modules' that cover different classroom-based problems common to ADHD: we anticipate that each child participant will receive two modules. SENCos will have the role of coordinating and supporting teachers and TAs to deliver the toolkit in conjunction with support from parents at home, and given agency over the duration, dose and discontinuation of strategies. Online supplemental figure 1 illustrates the participant structure and roles within a school (see online supplemental material).

## Intervention description

The Tools for Schools *FLEX* toolkit will primarily be a digital resource. The current logic model is shown in figure 2 and the toolkit outline is shown in the supplemental material (online supplemental figure 2). The key goals of the toolkit that will foster sustainable behaviour change are to:
1. *Understand* the child and their communications
2. *Adapt* the classroom and school expectations
3. *Support* the child to:

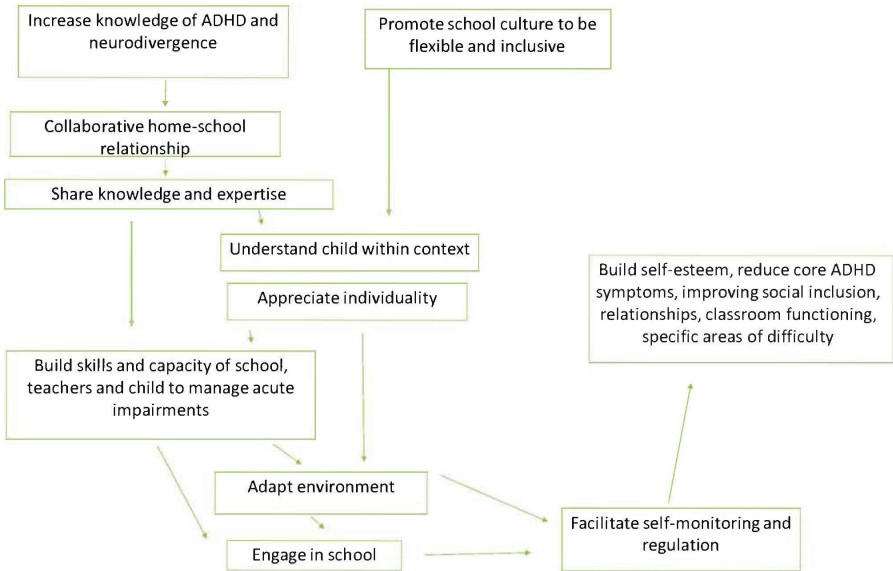

**Figure 2** Intervention draft logic model. ADHD, attention deficit/hyperactivity disorder.

A. reduce acute impairments

B. increase skills

C. improve self-identity through providing strategies that change the environment and expectations around the child

This results in the child being able to consciously monitor and self-regulate.

4. *Sustain* and adapt across time.

Table 3 shows the data collection schedule relative to the stage within the study. A full intervention description can be found in the online supplemental material.

## Qualitative data collection

### Toolkit iteration feedback

The toolkit will go through several iterations, refined based on ongoing data collection and analysis. Flexibility allowing for this is a strength of the study design.[31] Brief semistructured 'feedback sessions' will be held by phone or in person with all participating school staff and parents during the intervention period, in order to understand where modifications need to be made to the design, content and measures of the intervention. Example questions from the topic guide for teachers are included in the online supplemental material. All changes made from the original prototype of the toolkit will be recorded in a change log, including the reason for the change, when the change was made and what aspects of the toolkit the change relates to. The planning group and the study advisory group will convene to discuss feedback or major suggested modifications, and to review minor changes in the change log.

### Process evaluation

During the 10-week follow-up period, detailed qualitative data will be collected to assess acceptability, perceived usefulness, fidelity and implementation of the intervention, exploring in more depth what participants did and did not do and explanations of their actions, with questions similar to those shown above. Focus groups comprising the constellation of all school staff involved with one child will be held for all participants. This format is familiar to school staff as professional discussions often take a 'case conference' format. Interviews by phone or an online platform such as zoom will be conducted with a purposive sample of half of the parents, using the same criteria detailed above (ie, at least 50% from areas with a deprivation index above the regional mean) and aiming to attain a balance of male and female parents. Children will take part in 'active interviews' during the process evaluation in the follow-up period with a researcher (at home or at school), where they will have an informal conversation about their experiences of the study, if they recalled any new things their teacher tried with them, how they felt about their behaviour targets (if they knew about them), and how they found the research process.

All children will be invited to an activity- based interview with the option of a paired interview, where they can bring an accompanying family member of their choosing.

Paired interviews can reduce stress, mitigate child protection risks and reduce issues of running a focus group where children in the school with traits of ADHD are identified to one another. The activity interviews will be designed with the planning group and piloted with children with ADHD, for example, these may involve collecting objects of different colours or taking photos around the school to represent answers to questions (such as *can you show me where you like to go if you feel you need to calm down? What are your favourite things about school? What do you not like about school?*), and using these objects or photos as prompts to discuss thoughts.

All participants in the process evaluation will be asked what they think the most important outcome would be for a future hypothetical trial of the toolkit, and why.

### Adverse effects of intervention

It is possible that the intervention will lead to adverse effects or negative impacts on participating children, such as feeling singled out due to the intervention leading to social exclusion from the peer group.[32] The study planning group will contribute to identifying potential adverse effects (*as of 21 March social exclusion is the most prominent concern*), and qualitative data will be collected from participants at the feedback and process evaluation stages to assess perceptions of negative impacts of the intervention. Modifications will be made as needed in relation to this through consultation with the planning group and the academic advisory team.

### Quantitative measures

There are two quantitative outcome measures that will be assessed in order to: (1) Determine which would be most suitable as the primary outcome in a future trial and (2) Evaluate whether they make sense and are relevant to those completing the measures. *ADHD symptoms* will be measured using the Strengths and Weaknesses of ADHD Symptoms and Normal-behavior Questionnaire, based on preferences of the planning group who considered several measures of ADHD symptoms,[33 34] reported by teachers; and *classroom functioning problem behaviour* subscale measured using the Social Skills Improvement System (SSIS),[35] that captures social, academic and competing problem behaviours in the classroom environment, reported by parents and teachers. These will be measured every 2 weeks during the baseline and intervention periods and every 3 weeks during the follow-up period. All planned quantitative measures are detailed in a table in the online supplemental material.

Other quantitative measures will be captured to assess whether there is indication of some improvement in the key areas the toolkit aims to target. Measures relating to children include *child dimensional psychopathology*, assessed using the teacher and parent-reported Strengths and Difficulties Questionnaire,[30] and *child-reported satisfaction with school* using the How I Feel About My School measure.[36] *Teacher well-being* will be measured using the Warwick-Edinburgh Mental Wellbeing Scale 14-item

**Table 3** Data collection schedule and toolkit implementation timeline

| School terms | Baseline term | | | | | | | | | | | | | Intervention term | | | | | | | | | | | | | Follow-up term | | | | | | | | | | | | |
|---|---|---|---|---|---|---|---|---|---|---|---|---|---|---|---|---|---|---|---|---|---|---|---|---|---|---|---|---|---|---|---|---|---|---|---|---|---|---|---|
| Week | 1 | 2 | 3 | 4 | 5 | 6 | 7 | 8 | 9 | 10 | 11 | 12 | 13 | 1 | 2 | 3 | 4 | 5 | 6 | 7 | 8 | 9 | 10 | 11 | 12 | 13 | 1 | 2 | 3 | 4 | 5 | 6 | 7 | 8 | 9 | 10 | 11 | 12 | 13 |
| **Toolkit steps** | | | | | | | | | | | | | | | | | | | | | | | | | | | | | | | | | | | | | | | |
| Know ADHD | | | | | | | | | | | X | X | X | | | | | | | | | | | | | | | | | | | | | | | | | | |
| Know me | | | | | | | | | | | X | X | X | | | | | | | | | | | | | | | | | | | | | | | | | | |
| Understand motivations and develop targets | | | | | | | | | | | | | X | X | X | | | | | | | | | | | | | | | | | | | | | | | | |
| Establish digital school-home notebook | | | | | | | | | | | | | | X | X | | | | | | | | | | | | | | | | | | | | | | | | |
| Map targets to modules and choose strategies | | | | | | | | | | | | | | X | X | | | | | | | | | | | | | | | | | | | | | | | | |
| Implement strategies (module 1) | | | | | | | | | | | | | | | | X | X | X | X | X | | | | | | | | | | | | | | | | | | | |
| Implement strategies (module 2) | | | | | | | | | | | | | | | | | | | | | X | X | X | X | X | X | | | | | | | | | | | | | |
| Study measures (data collection) | | | | | | | | | | | | | | | | | | | | | | | | | | | | | | | | | | | | | | | |
| Measures of ADHD symptoms | X | X | X | X | X | X | X | X | X | X | X | X | X | X | X | X | X | X | X | X | X | X | X | X | X | X | X | | | X | | | X | | | X | | | X |
| Measures of classroom functioning | X | X | X | X | X | X | X | X | X | X | X | X | X | X | X | X | X | X | X | X | X | X | X | X | X | X | X | | | X | | | X | | | X | | | X |
| Toolkit iteration feedback (qualitative) | | | | | | | | | | | | | | | | | | X | | | | | | | | X | | | | | | | | | | | | | |
| Process evaluation: fidelity, acceptability, perceived usefulness | | | | | | | | | | | | | | | | | | | | | | | | | | | X | X | X | X | X | X | X | X | X | | | | |
| Child dimensional psychopathology and school functioning | X | | | | | | | | | | | | | | | | | | | | | | | | X | | | | | | | | | | | | X | | |

**Table 3** Continued

| School terms | Baseline term | | | | | | | | | | | | | Intervention term | | | | | | | | | | | | | Follow-up term | | | | | | | | | | | | |
|---|---|---|---|---|---|---|---|---|---|---|---|---|---|---|---|---|---|---|---|---|---|---|---|---|---|---|---|---|---|---|---|---|---|---|---|---|---|---|---|
| Week | 1 | 2 | 3 | 4 | 5 | 6 | 7 | 8 | 9 | 10 | 11 | 12 | 13 | 1 | 2 | 3 | 4 | 5 | 6 | 7 | 8 | 9 | 10 | 11 | 12 | 13 | 1 | 2 | 3 | 4 | 5 | 6 | 7 | 8 | 9 | 10 | 11 | 12 | 13 |
| Child-reported satisfaction with school | X | | | | | | | | | | | | | | | | | | X | | | | | | | X | | | | | | X | | | | | | | X |
| Teacher well-being | X | | | | | | | | | | | | | | | | | | X | | | | | | | X | | | | | | X | | | | | | | X |
| Healthcare and education resource use | X | | | | | | | | | | | | | | | | | | | | | | | | | X | X | | | | | | | | | | | | X |
| Child quality of life | X | | | | | | | | | | | | | | | | | | | | | | | | | X | X | | | | | | | | | | | | X |

Online supplemental material contains information on the Toolkit steps. Measures are: ADHD symptoms: the Strengths and Weaknesses of ADHD and Normal Behavior Scale (SWAN); Classroom functioning: problem behaviour subscale of the Social Skills Improvement System (SSIS); Dimensional psychopathology and school functioning: the Strengths and Difficulties Questionnaire and the academic competence and social skills subscales of the SSIS; Satisfaction with school: the How I Feel About My School Questionnaire; Teacher well-being: the Warwick-Edinburgh Mental Well-Being Scale 14-item Teacher Survey, and the Relationship with Work survey from the Maslach Burnout Inventory- General Survey, and the Teacher Sense of Efficacy Scale; healthcare and education resource use is a bespoke questionnaire available from the authors on request; and Child quality of life is measured by child-report and parent proxy Child Health Utility-9D.

ADHD, attention deficit/hyperactivity disorder.

Teacher Survey[37] the Teachers' Sense of Efficacy Scale,[38] and the Relationship with Work Survey from the Maslach Burnout Inventory-General Survey,[39 40] and the social skills and academic competence subscales from the SSIS will be completed by teachers. These will be collected at the beginning and end of the baseline period, at the end of the intervention term and at the end of the follow-up term. Other module-specific measures which are not covered by the above measures will be administered depending on the modules used by each child, for example, the Harter Self-Perception Profile for Children to measure self-esteem.[41]

Methods for collecting data on *healthcare and education resource use* and *health-related quality of life* will also be established and tested. Child quality of life (child-report and parent-proxy) will be measured using the Child Health Utility for Economic Evaluation (CHU9D).[42] A tool to accurately capture education and healthcare resource use will be developed for the study, drawing on the Client Service Receipt Inventory[43] and measures in the Database of Instruments for Resource Use Management Repository. Participants will complete this at baseline and follow-up, and provide qualitative information on changes in resource use during the follow-up process evaluation. A data collection form will also capture data on resource use and costs associated with the set-up and delivery of the intervention.

*Observational measures of child behaviour* in the classroom will also be tested throughout the study. These include but are not limited to the Classroom Observation Code,[44] and the Teacher-Pupil Observation Tool.[45] Observations will be carried out at school by members of the research team, trained to an acceptable reliability (0.7 or higher).

All data will be collected online via secure servers, or on paper, and manual data entry will be double checked.

### Analysis

#### Aim 1: To assess whether the toolkit is feasible and acceptable for schools to implement

We will use the definition of acceptability proposed by Sekhon *et al*[46]: 'a multi-faceted construct that reflects the extent to which people delivering or receiving a healthcare intervention consider it to be appropriate, based on anticipated or experienced cognitive and emotional responses to the intervention'. Qualitative data from both the toolkit iteration interviews and the process evaluation interviews and focus groups will be audio recorded, transcribed and analysed using the Framework method and Thematic analysis.[47] An initial coding framework will be developed after reading and re-reading transcripts. Deductive codes derived from the research questions and inductive codes within the data will be identified. This framework will be applied independently to three transcripts; discrepancies in coding will be discussed and revisions made to the framework which will then be applied to the remaining transcripts, with further discussion and revision where necessary.

Responses relating to each component of the toolkit will be synthesised in a framework in order to assess the acceptability of the evolving and final prototype; specifically to assess whether the components and the overall toolkit are considered appropriate to support children with ADHD in school. In addition, acceptability of the toolkit will be quantitatively assessed against the study progression criteria that relate to completion of each component of the toolkit, attendance at toolkit-related meetings, and the percentage of occasions the participant reports using the toolkit as instructed. The qualitative findings will be used to explain and further explore quantitative results in a mixed-methods synthesis. Modifications will be made iteratively as findings emerge, and by the end of the study each indicator should be in the 'green' range in order to indicate that the toolkit is acceptable.

### Aim 2: To establish whether a future trial of effectiveness is feasible

Data on the research process and participant compliance will be evaluated. The quantitative data on recruitment and retention of participants, completion of measures by each participant group (parents, child, teacher, observed) and adherence to the intervention protocol will be assessed relative to the relevant progression criteria described in table 2. These have been informed by existing school-based trials.[48 49] As above, qualitative findings from the process evaluation and toolkit iteration feedback will be integrated to explain quantitative findings. Should all relevant progression criteria be rated 'green', a future trial of ef will be considered feasible. Should criteria be amber or red as the study progresses, potential mitigating modifications will be discussed, such as removing the requirement for parents to actively engage with the toolkit.

### Subaim A

In order to identify suitable outcome measures to assess core ADHD symptoms, child and teacher well-being and an appropriate primary outcome for a future definitive trial, the acceptability and feasibility of using the individual measures will be evaluated following the steps above and in relation to the threshold set in the progression criteria. Measures will be considered suitable if there is a high level of compliance with completion (ie, each meet the green criteria), and qualitative findings indicate no major barriers to completing the measure on repeated occasions. Should measures be identified as unsuitable between toolkit iterations, new measures will be identified with the planning group and the data collection plan modified. For example, if the ADHD symptoms measure is too burdensome to complete as frequently as required, other briefer ADHD symptom scales will be explored, or the data collection frequency may be modified. The two versions of the CHU9D will be compared to assess whether child-report is sufficient. To *identify an appropriate primary outcome for a future trial*, participants' qualitative responses to direct questions regarding what they

think the most important outcome should be (and why), collected during the process evaluation, will be synthesised. Findings will be discussed with the planning group and advisory team prior to deciding on the final primary outcome measure.

### Subaim B

In order to test a framework for costing the toolkit, and for assessing cost-effectiveness in a future definitive trial, the detailed information collected from the draft framework will be assessed following baseline data collection, and modifications made to the framework where free-text responses indicate additional resources not captured in the draft framework. The process evaluation interviews and focus group data will also contribute to understanding changes in resource use, which will be aligned with the follow-up responses regarding resource use in order to ascertain whether the framework is adequate and sensitive to changes in resource use.

The data captured on a bespoke form relating to the costs of setting up and delivering the toolkit will be assessed to expand the framework for accurately costing the toolkit's delivery and potential savings, and the costs of delivering the prototype intervention will be calculated (including costs for staff time, training and delivery of components). This framework will be revised in line with study findings with the aid of health economists, with a final framework developed by the end of the study that would allow for cost-effectiveness to be formally assessed in a future definitive trial.

### Subaim C

Assess whether measures of behaviour, classroom functioning and teacher-reported ADHD symptoms indicate improvement following use of the toolkit. *To explore whether there is indication that the toolkit may be efficacious*, quantitative data will be visually analysed within schools and participants, describing the nature of change over time and in relation to the intervention, following recommendations.[31] Non-parametric analysis of the data will be considered, for example, by calculating Tau-U statistics. As the toolkit will be refined between schools participating, it would be irrelevant to compare measures across individuals who received different iterations of the toolkit; the focus will therefore be on assessing whether there is initial indication of improvement in outcomes, and whether any effect is replicated across individuals. Comparisons between the magnitude of change seen between those who receive earlier and later iterations of the toolkit will be carried out as it is anticipated that later iterations, modified to be more acceptable and feasible, would result in greater evidence of behaviour change. This will be an exploratory analysis. The core and target behaviours chosen across study participants will also be explored in a descriptive manner in order to assess the use of the behaviour web, targets and modules.

### Missing, unused and spurious data, and inclusion in analysis

All data will be used in analysis. Patterns of missing quantitative data will be evaluated as part of assessing the feasibility and acceptability of the intervention. Missing data could relate to the feasibility and acceptability of the intervention, or the utility and acceptability of the measure itself (i.e., Aim 1 and Subaim A), and so through the qualitative data collection we will aim to elicit how patterns of missing data relate to our aims and make modifications accordingly.

## ETHICS AND DISSEMINATION
### Ethics and governance approval

Ethical approval has been awarded by the University of Exeter College of Medicine and Health Research Ethics Committee (Ref Jan22/B/300). Informed consent will be sought from all adult participants (and by parents for child participants; an example information sheet and consent form are provided in the online supplemental material). Children will provide assent and ongoing assent will be judged by the research team throughout. Should it become clear that a child does not want to continue to participate in data collection or the study, withdrawal will be discussed with parents and school staff. Should a child express that they wish to stop during a single instance of data collection, this will be stopped for that day. The child's feelings will be respected and discussed with the child, their parents and school staff about their future involvement. Ultimately, decisions will be made in line with the best interests of the child. All data collected will comply with General Data Protection Regulations, being stored on secure servers and only accessible by the research team. Some data will be shared as part of the intervention, for example, teachers' ratings of ADHD symptoms will be viewable by the SENCo and express permission will be sought for any planned data sharing between participants. Participants will be asked whether their data can be made available following the study for future analysis or related research. Those that consent will contribute their data to a fully anonymised data set accessible to bona fide researchers. There are several operational issues and associated risks that are involved in performing this study. It is possible that the intervention will lead to adverse outcomes[32] such as study children feeling singled-out from their peers due to the intervention. The planning group will discuss anticipated potential adverse outcomes due to the intervention, and qualitative feedback will be sought from participants both during their use of the intervention and in the follow-up period. Further risks (aside from those detailed in progression criteria in table 2) have been identified and mitigation plans detailed, as well as key study limitations noted (see online supplemental material).

### Dissemination

Participants who consent to their anonymised data being shared for future research purposes will have their data archived in an open access data set. The results from the study will be presented at relevant national and international conferences (eg, the ADHD World Congress, Eunethydis, the Association for Child and Mental Health national conferences). Separate publications will describe: the iterative development process, the main outcomes of the case-series study, development of the resource use tool, and an in-depth qualitative analysis of data collected during the iteration and process evaluation; authorship will be determined using the International Committee of Medical Journal Editors (ICJME) criteria. A detailed dissemination plan and documents will be developed with the planning group and will include blogs and podcasts, recorded and live talks, and events and engagement activities to reach a broader audience of interested stakeholders and policymakers.

**Author affiliations**

[1]Children and Young People's Mental Health Research Collaboration (ChYMe), University of Exeter Medical School, University of Exeter, Exeter, UK

[2]Department of Psychology, University of Exeter, Exeter, UK

[3]Graduate School of Education, University of Exeter, Exeter, UK

[4]Population Health Sciences, Bristol Medical School, University of Bristol, Bristol, UK

[5]Institute of Psychiatry, Psychology and Neuroscience, King's College London, London, UK

[6]Department of Psychiatry, University of California San Francisco, San Francisco, California, USA

[7]Department of Psychiatry, University of Cambridge, Cambridge, UK

**Acknowledgements** The authors thank Suzie Holt, Eleanor Bryant, Becky Gudka and Charlotte Kelman of the Exeter University ChYMe (Children and Young People's Mental Health (ChYMe)) research collaboration for additional contribution. The authors also thank the members of the Tools for Schools planning group for their ongoing collaboration (https://blogs.exeter.ac.uk/toolsforschools/about-us/planning-group/).

**Contributors** AR, DM and TF conceptualised the work; AR, BD, DM, RH, ES-B, LP, TF, JK designed and drafted the research programme, led by AR. AR drafted the manuscript, and BD, DM, RH, ES-B, LP, TF and JK critically revised the final manuscript.

**Funding** This report is independent research supported by the National Institute for Health Research NIHR Advanced Fellowship - Stage 2, to AR (grant ref NIHR300591). The views expressed in this publication are those of the author(s) and not necessarily those of the NHS, the National Institute for Health Research or the Department of Health and Social Care. TF receives funding from Place2Be, a children's mental health charity (grant ref: N/A).

**Competing interests** None declared.

**Patient and public involvement** Patients and/or the public were involved in the design, or conduct, or reporting, or dissemination plans of this research. Refer to the Methods section for further details.

**Patient consent for publication** Not applicable.

**Provenance and peer review** Not commissioned; externally peer reviewed.

**ORCID iDs**
Abigail Emma Russell http://orcid.org/0000-0002-2903-6264
Judi Kidger http://orcid.org/0000-0002-1054-6758
Tamsin Ford http://orcid.org/0000-0001-5295-4904

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
