## [Reviewer comments · BMJ Open]

ARTICLE DETAILS

TITLE (PROVISIONAL)	Protocol for an iterative case-series study to investigate the feasibility and acceptability of a school-based intervention for children with traits of ADHD
AUTHORS	Russell, Abigail; Dunn, Barney; Hayes, Rachel; Moore, Darren; Kidger, Judi; Sonuga-Barke, Edmund; Pfiffner, Linda; Ford, Tamsin

VERSION 1 – REVIEW

REVIEWER	Sciberras, Emma Murdoch Childrens Research Institute, Centre for Community Child Health I am a clinical psychologist and researcher working in the area of ADHD. I conduct a number of research studies in the area of ADHD spanning both intervention and non-intervention studies. I receive research funding from the Australian National Health and Medical Research Council, the Medical Research Future Fund, the Waterloo Foundation and veski. I am an elected executive board member of the Australian ADHD Professionals Association.
REVIEW RETURNED	20-Sep-2022

GENERAL COMMENTS	Thank you for the opportunity to review the protocol for this study. The area of school-based interventions for ADHD sorely needs more evidence, and it is my opinion that this study will be an excellent addition to the literature. I appreciate the need for careful piloting of the intervention before moving to an efficacy trial, and it is fantastic to see a protocol based on this piloting process. Overall, relatively little research considers real life implementation challenges in the context of ADHD treatment research, and it refreshing to see this protocol focusing on this area. Some key strengths of the study are the consideration of teachers' views, lived experience input, the proposed iterative approach, the mixed methods design, the focus on feasibility and acceptability, multi-informant data collection, capture of both potential positive and adverse outcomes, and modular intervention design. The study has clear aims and objectives. I very much look forward to seeing the results from this study. I am particularly interested in the selected intervention components and have included some comments about these below. In particular, step 3, seems to have more of a focus on 'changing the child' as opposed to changing the environment. The intervention draft logic refers to 'adapt the environment' but I couldn't see where in the program this would occur. Will there be a component of the intervention focused on inclusive education strategies? It seems like a missed opportunity not to include this in Step 1. I did see that there will be general guidance for schools to plan diversity/inclusivity
--

events but these are reported to be optional. There is discussion of the importance of inclusion in step 5 but overall, I found it difficult to understand the balance of the intervention in terms of strategies focusing on inclusion vs strategies to change children's behaviour.

Please find some more specific comments regarding each section of the manuscript below:

Introduction:

- This section provides an excellent review of the literature and rationale for the study.
- It is important to add in referencing to support some of the points made on page 4, paragraph 1 e.g., 'Pharmacological treatment is available and effective for some, but not appropriate, acceptable or tolerable for all children', 'Response is often partial, with tolerance taking over 2-3 years to develop', 'Medication is not appropriate or preferable for all children with ADHD or as a first-line treatment for children with traits that are not severe...' I understand the point the authors are making here but believe that these statements should be supported by references. It is also important to ensure that this is balanced with the potential benefits of medication.
- Page 5 – top paragraph where limitations of existing school-based interventions are reported, requires referencing. Also provide a reference to the Intervention Mapping approach for those unfamiliar.

Figure 2 - I think this is an excellent addition to the paper. Check for typos.

Table 2 - I really like this table reporting on study progression criteria.

- Regarding recruitment & retention, will be >65% be considered in each group (e.g. schools, parents) separately, or overall? I assume this will be considered in the separate subgroups but this could be made clearer in the table.
- I was curious about the justification for only at least 50% watching the introduction video as being a green. Low initial engagement may well have flow on benefits to parents completing follow-up surveys over time (if parents initial video watching is around 51% would you still expect at least 50% of parents to complete follow-up surveys). I have similar thoughts about the threshold for the child strengths activity.
- I wonder about the low minimum threshold for parent completed surveys, I would have assumed that aiming for higher would be ideal.
- It is unclear from the table who will be completing observational measures (this is also unclear in the text) and again why a minimum of 50% is considered acceptable.
- The final row is unclear – is this overall follow-up measures completed or for particular reporters?

Table 3 – it would be really helpful for footnotes to be added so that the actual measures being used to assess any of the key constructs are clear e.g., how will child dimensional psychopathology be measured? This is reported in text but would be useful if clear in the table.

Supplementary materials

- Given that many parents in the study will not have ever considered ADHD and their child may not well meet criteria for ADHD, how will 'Step 1 – know ADHD' be modified for this group? Will it cover both

	subthreshold and full threshold ADHD? I am wondering about whether it will be confronting for parents of children with elevated symptoms to be learning about ADHD if they do not consider their child to have ADHD? - Step 3 – how much does the intervention focus on ABA therapy approaches? I am wondering about the lived experience input provided to date on these types of strategies? I also wondered about the behaviour target provided about staying in the seat. Is there an opportunity to think more broadly about things that would really enhance children’s school experiences and feelings of efficacy, and ways to draw upon existing strengths? I assume this is in the intervention and the example is just one of many but perhaps a broader set of examples will help the reader to get a feel for the intervention. Broader comments for consideration: - What will children be told about the intervention and why they have been selected for this intervention? Will children be included in the discussion of behaviour targets? - It is great to see the collection of parent and teacher reports, but I do think the design could be enhanced by increasing considering child views. I see that children will provide data on their quality of life, but it seems like a missed opportunity not to collect data from the children themselves about their experience of the intervention. It does appear that the children will be asked about important outcomes for a future trial, but I wasn’t clear from the description whether children will be given an opportunity to comment on their experiences of the intervention and in particular, the behaviour targets decided by parents and teachers. - Regarding children, I wondered whether there is a missed opportunity to collect data about other aspects of child wellbeing like child self-esteem? And children’s school engagement? Furthermore, I wondered whether some broader outcomes for teachers could be considered like teacher self-efficacy and knowledge and feelings of confidence/competence in ADHD? Thank you for the opportunity to learn more about your work.
--	--

REVIEWER	Shah, Ruchita Post Graduate Institute of Medical Education and Research
REVIEW RETURNED	20-Oct-2022

GENERAL COMMENTS	Overall, the manuscript is well-written and has provided adequate details. The study itself has several strengths; the most important one is that the process of development and preliminary assessment of feasibility and acceptability of the intervention is dynamic, inclusive of all stakeholders, iterative and self-corrective in nature. There are few points that may be addressed. Essential:  1. Clearly mention exclusion criteria 2. Clearly mention potential study limitations 3. The SPIRIT checklist in the Supplementary material is not complete. Desirable:  1. The authors have mentioned application of Intervention mapping approach. It will be helpful to the readers if steps within IM that will be followed in the index protocol are explicitly presented in the main draft briefly. 2. A table providing details of the potential methods/ scales/
--

	instruments selected for specific outcomes will help the reader 3. The composition of the planning group may be provided with the numbers from each group. 4. The authors state - All children will be invited to an activity- based interview with the option of a paired interview, where they can bring an accompanying family member of their choosing – Is this during process evaluation phase or baseline phase. This is not clear as this paragraph appears under process evaluation. 5. Will the qualitative interviews be audio-video recorded for later analysis? If so, mention that explicitly I look forward to the completion of this work and its reporting.
--	---

VERSION 1 – AUTHOR RESPONSE

Reviewer: 1

Dr. Emma Sciberras, Murdoch Childrens Research Institute

Comments to the Author:

Thank you for the opportunity to review the protocol for this study. The area of school-based interventions for ADHD sorely needs more evidence, and it is my opinion that this study will be an excellent addition to the literature. I appreciate the need for careful piloting of the intervention before moving to an efficacy trial, and it is fantastic to see a protocol based on this piloting process. Overall, relatively little research considers real life implementation challenges in the context of ADHD treatment research, and it refreshing to see this protocol focusing on this area. Some key strengths of the study are the consideration of teachers' views, lived experience input, the proposed iterative approach, the mixed methods design, the focus on feasibility and acceptability, multi-informant data collection, capture of both potential positive and adverse outcomes, and modular intervention design. The study has clear aims and objectives. I very much look forward to seeing the results from this study.

Response: Thank you for your positive appraisal of the need for and value of this study.

I am particularly interested in the selected intervention components and have included some comments about these below. In particular, step 3, seems to have more of a focus on 'changing the child' as opposed to changing the environment. The intervention draft logic refers to 'adapt the environment' but I couldn't see where in the program this would occur. Will there be a component of the intervention focused on inclusive education strategies? It seems like a missed opportunity not to include this in Step 1. I did see that there will be general guidance for schools to plan diversity/inclusivity events but these are reported to be optional. There is discussion of the importance of inclusion in step 5 but overall, I found it difficult to understand the balance of the intervention in terms of strategies focusing on inclusion vs strategies to change children's behaviour.

Response: This is a really good point, as we have now made much more progress with developing the strategies it has become easier to clarify this coherently. We have added to the supplementary material for step 3:

“This step of the intervention focusses on what achievements it is hoped the child will make, however the strategies that will be implemented to achieve these are primarily designed to adapt the environment around the child, for example scaffolding them with classroom tools and activities to improve organisation through making it clearer to the child where their possessions can be found or visual aids to demonstrate what will be needed for the next activity; introduction of whole-class games on identifying and providing positive feedback to peers where all classmates are reinforced for “picking up the positives” to improve self-

esteem; incorporation of movement-based learning strategies throughout the school day to improve ability to concentrate, and the teacher changing the format of the work that the child is asked to complete.”

We have also added further detail to step 6:

“Strategies have been designed to be primarily group-based games or activities, or reminders and support for teachers to incorporate new techniques to adapt around the child, with the aim that this will lead to improved outcomes for the child. None of the strategies require the child to be removed from the classroom, and although some could be used 1:1 with the child, they are primarily presented as group-level activities to promote inclusion. A significant number of strategies focus on “belonging and relating”, specifically aiming to promote peer inclusion, celebrate individual differences, and destigmatise neurodevelopmental (and other) differences.”

And regarding the SLT inclusion challenge, we have expanded further on the reason this is not mandatory in step 1:

“The toolkit provides step-by-step instructions for the SLT to conduct an “Inclusion challenge” and makes suggestions as to what systems and processes within the school they may consider changing as part of this.”

and

“Given the aim of this study is to assess feasibility and acceptability of implementation in a real-world context, we have made the decision not to make any component of the flex toolkit mandatory (although we strongly suggest that staff complete the ADHD training and the SLT do the inclusion challenge); the current UK school context is facing severe pressures and resource constraints, and we anticipate that uptake of components will be better and more representative of naturalistic use of the toolkit if they are not mandated. Whether or not to mandate this component in a future trial will be considered based on the findings of this study.”

Please find some more specific comments regarding each section of the manuscript below:

Introduction:

- This section provides an excellent review of the literature and rationale for the study.

Response: Thank you.

- It is important to add in referencing to support some of the points made on page 4, paragraph 1 e.g., ‘Pharmacological treatment is available and effective for some, but not appropriate, acceptable or tolerable for all children’, ‘Response is often partial, with tolerance taking over 2-3 years to develop’, ‘Medication is not appropriate or preferable for all children with ADHD or as a first-line treatment for children with traits that are not severe...’ I understand the point the authors are making here but believe that these statements should be supported by references. It is also important to ensure that this is balanced with the potential benefits of medication.

Response: Thank you for highlighting this. We have added relevant citations and additional detail on the potential benefits of medication to the introduction.

- Page 5 – top paragraph where limitations of existing school-based interventions are reported, requires referencing. Also provide a reference to the Intervention Mapping approach for those unfamiliar.

Response: Thank you for highlighting this. We have added the relevant citation and some brief further detail to the introduction.

Figure 2 - I think this is an excellent addition to the paper. Check for typos.

Response: Thank you for highlighting this. We have fixed the typos.

Table 2 - I really like this table reporting on study progression criteria.

- Regarding recruitment & retention, will be >65% be considered in each group (e.g. schools, parents) separately, or overall? I assume this will be considered in the separate subgroups but this could be made clearer in the table.

Response: You are correct, we have clarified this within the table.

- I was curious about the justification for only at least 50% watching the introduction video as being a green. Low initial engagement may well have flow on benefits to parents completing follow-up surveys over time (if parents initial video watching is around 51% would you still expect at least 50% of parents to complete follow-up surveys). I have similar thoughts about the threshold for the child strengths activity.

Response: Thank you for highlighting this. As we added in the intervention description step 1 detailed above, we are interested in what components of the toolkit are useable and acceptable, and think we will learn important lessons from what is taken up or not. We therefore do not want to judge study success by these metrics, as it may be (for example) that individual strategies are judged to be extremely helpful but the introduction and child strengths activities too burdensome in their current form. If we were to increase the threshold of the continuation criteria for these (which will be used to judge progression to a trial as a whole across all criteria) then we run the risk of losing valuable information as to why these were not watched or engaged with- however if uptake were to be as low as 50% we would certainly investigate why that was and aim to improve it during the course of the study.

Regarding the parent video component, the toolkit is being designed to be useful primarily within schools, and parent active engagement is therefore optional for parents who are in a situation where they are able to engage, hence the particularly low threshold for this criterion.

Regarding the child strengths activity, many schools have told us that they do a similar exercise to this at the beginning of each school year, so it may well be that an existing activity or knowledge is adapted to meet the aims of this activity, again explaining the low threshold.

- I wonder about the low minimum threshold for parent completed surveys, I would have assumed that aiming for higher would be ideal.

Response: Please see our point above about this being primarily a school-based intervention, with impacts on child behaviour most likely to be proximal to school and potentially undetectable at home. Should the completion rate really be this low, it would indicate that a future study with solely school-reported measures would be more feasible than the model we are proposing in the feasibility study. We have also made our decision informed by previous RCTs in similar settings, where parent-completed follow-up measures tend to be around 50% (e.g. the STARS trial; Ford et al 2019)

- It is unclear from the table who will be completing observational measures (this is also unclear in the text) and again why a minimum of 50% is considered acceptable.

Response: Thank you for bringing this to our attention. We have added detail: observations are being carried out by members of the research team, trained to achieve acceptable reliability (0.7 or higher) on the selected measure.

On review, we do agree that aiming to achieve higher rates of observations would be desirable and important, so we have amended the figures in the table to Green: >70%, Amber 50-70%, and Red <50%.

- The final row is unclear – is this overall follow-up measures completed or for particular reporters?
Response: Thank you for highlighting this. We have clarified that by this, we mean follow-up qualitative data collection, and that the measure is for each type of respondent (school staff, parents, child) rather than the cumulative number.

Table 3 – it would be really helpful for footnotes to be added so that the actual measures being used to assess any of the key constructs are clear e.g., how will child dimensional psychopathology be measured? This is reported in text but would be useful if clear in the table.

Response: We have added the measures to the footnote of table 3.

Supplementary materials

- Given that many parents in the study will not have ever considered ADHD and their child may not well meet criteria for ADHD, how will 'Step 1 – know ADHD' be modified for this group? Will it cover both subthreshold and full threshold ADHD? I am wondering about whether it will be confronting for parents of children with elevated symptoms to be learning about ADHD if they do not consider their child to have ADHD?

Response: This is a really important point. The step 1 learning is titled “know traits of ADHD” and uses minimal language about ADHD as a disorder, and talks more about traits of poor attention or concentration, impulsive behaviour or acting without thinking, and hyperactivity. In the learning materials we include a segment on how this relates to the label or diagnosable disorder of ADHD. We are interested to obtain feedback from parents about this.

All of the parents in the study have had contact with their child’s SENCo about the child’s difficulties or perceived need for additional support in school prior to the study. When selecting and approaching parents, this is discussed expressly with the SENCo and they have a discussion with parents prior to the research team becoming involved. The study information sheet also talks about traits of ADHD, and parents are therefore aware of the nature of the toolkit before they get to Step 1.

We have added this detail to the Step 1 description in the supplementary materials.

- Step 3 – how much does the intervention focus on ABA therapy approaches? I am wondering about the lived experience input provided to date on these types of strategies? I also wondered about the behaviour target provided about staying in the seat. Is there an opportunity to think more broadly about things that would really enhance children’s school experiences and feelings of efficacy, and ways to draw upon existing strengths? I assume this is in the intervention and the example is just one of many but perhaps a broader set of examples will help the reader to get a feel for the intervention.

Response: Thank you for this comment. The intervention primarily uses theoretical approaches around behaviour change and related methods taxonomies, rather than being grounded in ABA. However, there is a lot of overlap in the methods that would be used in ABA that are relevant for children in this study as they are both based on behaviourist principles and theories of learning.

The A-B-C analysis being used in Step 3 to understand motivations and develop targets is included with the purpose of encouraging school staff and parents to explicitly consider the various contexts around the child and their behaviour, highlighting targets for intervention that are not intrinsic to the child (we have added this to the Supplementary material).

The assumptions you detail above are accurate- there are various targets for the toolkit that would potentially improve peer relationships, self-esteem and other experiences of school. However, the specific goals and steps towards targets that may be formalised for the child would be more specific than these more diffuse benefits to hopefully capture incremental improvement in a manner that teachers, parents and the child can see progression in. Now we have made more progress with the intervention development we have added some of these to the example (“having more than one friend to play with at lunchtime, starting with one day per week and increasing over time; having three adults at school they can go to if they have a problem, starting with one adult and knowing what and how to communicate to them if there is a problem”). The repeated quantitative measures in the study would hopefully capture the nature and extent of broader benefits.

Regarding the lived experience input provided, this has been extensive. We have added the below to the supplementary material (unfortunately journal word count does not allow us to include this in the main text), and referred to this in the main text. The process has been:

“-Parents, children with ADHD, adults with ADHD, SENCOs, educational psychologists and teachers (the planning group) were initially informally interviewed about their experiences, and what they perceive works for them. These findings were narratively synthesised and written up in one document.

-They contributed to an exhaustive list of the kinds of behaviours that may form targets for the intervention, and the modules that these would fall within (e.g. being ‘easily led by others’ mapping to the ‘pausing for thought’ module); with some behaviours mapping to multiple modules depending on the underlying reason (e.g. ‘risky or dangerous behaviour’ mapping to either ‘paying attention and engaging’ or ‘pausing for thought’ depending on whether the primary reason is impulsivity or inattentiveness).

-The intervention logic model of change was reviewed and refined with the planning group, who then had input into the performance objectives (the smallest steps of behaviour change required to meet each behavioural goal for each person e.g. teacher, child, senior leadership)

through reviewing the lists in an 'activity workbook' format, adding thoughts and attending discussion meetings.

-These performance objectives were also categorised by the determinants of behaviour (whether the change in behaviour required aligned with skills, values, beliefs, attitudes, knowledge or experience). Initial strategy ideas were then drafted by AR to match performance objectives with input from behaviour change taxonomies to select relevant delivery methods.

-These draft strategy ideas were then reviewed by the planning group, and changed entirely or improved based on their input. Strategy instructions were then written by AR, reviewed by the planning group once more and refined and the prototype toolkit was produced. PDF resources to support teachers in delivering each strategy were generated by the research team, with the planning group suggesting resource ideas and providing further input and feedback at the 'alpha testing' stage of the prototype. Resources and strategies were then refined and finalised, prior to the beta version (to be used in the feasibility study) being rolled out."

Broader comments for consideration:

- What will children be told about the intervention and why they have been selected for this intervention? Will children be included in the discussion of behaviour targets?

Response: We have added the following to the methods: "Children will be informed of the broad purpose of the research through an age-appropriate information sheet and assent form, unless their parents request otherwise." And to the supplementary information (Step 3), that implies the child will be involved by default: ", and parents will make the decision as to whether they wish for their child to be involved in these meetings".

- It is great to see the collection of parent and teacher reports, but I do think the design could be enhanced by increasing considering child views. I see that children will provide data on their quality of life, but it seems like a missed opportunity not to collect data from the children themselves about their experience of the intervention. It does appear that the children will be asked about important outcomes for a future trial, but I wasn't clear from the description whether children will be given an opportunity to comment on their experiences of the intervention and in particular, the behaviour targets decided by parents and teachers.

Response: Thank you for this comment. We have added additional detail about the child interviews that form part of the process evaluation (we completely agree with the reviewers' point!). We have added this in:

"Children will take part in 'active interviews' with a researcher (at home or at school), where they will do an activity they choose and have an informal conversation about their experiences of the study, if they recalled any new things their teacher tried with them, how they felt about their behaviour targets (if they knew about them), and how they found the research process."

In addition, children are completing a repeated measure of satisfaction with school: the "How I feel about my school" questionnaire as well as the quality of life measure. This is already detailed in the quantitative measures section.

- Regarding children, I wondered whether there is a missed opportunity to collect data about other aspects of child wellbeing like child self-esteem? And children's school engagement? Furthermore, I wondered whether some broader outcomes for teachers could be considered like teacher self-efficacy and knowledge and feelings of confidence/competence in ADHD?

Response: Please see above regarding the HIFAMS questionnaire (school engagement). Other measures will be captured depending on the module the teacher selects, if they are using the "feeling good" or "belong and relating" module then there will be additional measures about self-esteem or peer belonging collected from the children when we complete the regular measures. We are using the Harter Self-Perception profile for children for child-reported self-esteem (Harter, 2012). We have added this detail to the measures section, as well as a table in the Supplementary material of all the quantitative measures.

Regarding teacher measures, the original protocol reported we would collect the Maslach Burnout Inventory, this is the Relationship with Work Survey component of the scale (detail added to protocol), and we are now also using the Teachers' Sense of Efficacy Scale (short form; Tschannen-Moran and Hoy, 2001) that we hope captures the broader outcomes mentioned above. Given the burden of data collection for teachers we have not included an ADHD-specific measure at this stage.

Thank you for the opportunity to learn more about your work.

Reviewer: 2

Dr. Ruchita Shah, Post Graduate Institute of Medical Education and Research

Comments to the Author:

Overall, the manuscript is well-written and has provided adequate details. The study itself has several strengths; the most important one is that the process of development and preliminary assessment of feasibility and acceptability of the intervention is dynamic, inclusive of all stakeholders, iterative and self-corrective in nature.

Response: Thank you for this positive review of our protocol.

There are few points that may be addressed.

Essential:

1. Clearly mention exclusion criteria

Response: Thank you. We have added to the recruitment section "There are no additional exclusion criteria".

2. Clearly mention potential study limitations

Response: Thank you for mentioning this. Some limitations are captured in the Article summary. We were unclear where to add additional limitations in this protocol, and the SPIRIT checklist (that we have followed for reporting) does not include a heading on this. We have

added a note in the ethics section that key study limitations are detailed in the Supplementary material, and added the following to the Supplementary material:

“This study is a feasibility study, and due to the iterative case-series design not all participants will get the same version of the study prototype. Therefore we will not be able to quantitatively measure or compare changes in our outcome measures across all study participants, instead we are restricted to examining whether individual trends of change are replicated across individuals.

In addition, due to the modular nature of the toolkit, the sample size for a future cluster randomised controlled trial may have to be large in order to obtain sufficient numbers of participants using each module.”

3. The SPIRIT checklist in the Supplementary material is not complete.

Response: Thank you for highlighting this. We have completed the checklist.

Desirable:

1. The authors have mentioned application of Intervention mapping approach. It will be helpful to the readers if steps within IM that will be followed in the index protocol are explicitly presented in the main draft briefly.

Response: Thank you for highlighting this. We have added the following to the introduction, and some further detail is also now provided in the ‘co-production’ subheading in the supplementary materials:

“Intervention Mapping has six steps, and the full process of the toolkit development will be detailed in a subsequent publication, however in brief these are: 1. Creating a logic model of the problem; 2. Defining programme outcomes and objectives and a logic model of change; 3. Programme design; 4. Programme production; 5. Programme implementation plan; and 6. Evaluation plan (Bartholomew et al., 1998).”

2. A table providing details of the potential methods/ scales/ instruments selected for specific outcomes will help the reader

Response: Thank you for this point. We have added a table of measures for each outcome to the Supplementary material.

3. The composition of the planning group may be provided with the numbers from each group.

Thank you for suggesting this. We have added this with [n=] to the relevant line where the planning group composition is described.

4. The authors state - All children will be invited to an activity- based interview with the option of a paired interview, where they can bring an accompanying family member of their choosing – Is this during process evaluation phase or baseline phase. This is not clear as this paragraph appears under process evaluation.

Response: This is during the follow-up term in the process evaluation. We have added this detail now.

5. Will the qualitative interviews be audio-video recorded for later analysis? If so, mention that explicitly

Response: Thank you, we have added this detail to the analysis section.

I look forward to the completion of this work and its reporting.

VERSION 2 – REVIEW

REVIEWER	Shah, Ruchita Post Graduate Institute of Medical Education and Research
REVIEW RETURNED	09-Dec-2022
GENERAL COMMENTS	Authors have satisfactorily responded to all the queries and suggestions. I wish them success and look forward to related publications as well as final results of this study.